# Regulatory Policies for Alcohol, other Psychoactive Substances and Addictive Behaviours: The Role of Level of Use and Potency. A Systematic Review

**DOI:** 10.3390/ijerph16193749

**Published:** 2019-10-04

**Authors:** Jürgen Rehm, Jean-François Crépault, Omer S.M. Hasan, Dirk W. Lachenmeier, Robin Room, Bundit Sornpaisarn

**Affiliations:** 1Institute for Mental Health Policy Research, Centre for Addiction and Mental Health, 33 Russell Street, Toronto, ON M5S 2S1, Canada; JeanFrancois.Crepault@camh.ca (J.-F.C.); Bundit.Sornpaisarn@camh.ca (B.S.); 2Campbell Family Mental Health Research Institute, Centre for Addiction and Mental Health, 33 Russell Street, Toronto, ON M5S 2S1, Canada; 3Dalla Lana School of Public Health, University of Toronto, 155 College Street, 6th floor, Toronto, ON M5T 3M7, Canada; 4Faculty of Medicine, Institute of Medical Science, University of Toronto, Medical Sciences Building, 1 King’s College Circle, Room 2374, Toronto, ON M5S 1A8, Canada; 5Department of Psychiatry, University of Toronto, 250 College Street, 8th floor, Toronto, ON M5T 1R8, Canada; 6Department of International Health Projects, Institute for Leadership and Health Management, I.M. Sechenov First Moscow State Medical University, Trubetskaya str., 8, b. 2, 119992 Moscow, Russia; 7Institute of Clinical Psychology and Psychotherapy & Center of Clinical Epidemiology and Longitudinal Studies (CELOS), Technische Universität Dresden, Chemnitzer Str. 46, 01187 Dresden, Germany; 8Chemisches und Veterinäruntersuchungsamt (CVUA) Karlsruhe, Weissenburger Strasse 3, 76187 Karlsruhe, Germany; 9Centre for Alcohol Policy Research, La Trobe University, Bundoora, Victoria 3086, Australia; R.Room@latrobe.edu.au; 10Centre for Social Research on Alcohol and Drugs, Department of Public Health Sciences, Stockholm University, 106 91 Stockholm, Sweden

**Keywords:** alcohol, policy, regulation, tobacco, opioids, cannabis, gambling, potency, use disorders

## Abstract

The object of this contribution based on a systematic review of the literature is to examine to what degree the level of use and potency play a role in regulatory policies for alcohol, other psychoactive substances and gambling, and whether there is an evidence base for this role. Level of use is usually defined around a behavioural pattern of the user (for example, cigarettes smoked per day, or average ethanol use in grams per day), while potency is defined as a property or characteristic of the substance. For all substances examined (alcohol, tobacco, opioids, cannabis) and gambling, both dimensions were taken into consideration in the formulation of most regulatory policies. However, the associations between both dimensions and regulatory policies were not systematic, and not always based on evidence. Future improvements are suggested.

## 1. Introduction

In contemporary society, the use of psychoactive substances is controlled—by regulating or prohibiting them [1]—because of their substantial public health impact [2,3,4]. Part of this impact is due to the addictive potential of these substances [5,6]. This contribution will analyse the policies used to regulate substance use ([1], specifically for alcohol [7], tobacco [8], and illegal drugs: [9]), focussing on the roles of level of use and potency underlying such regulations. In doing so, we realize that regulations are clearly not based on only these two dimensions, nor do we make a judgement that they should be. The objective of this contribution is rather to examine to what degree the level of use and potency play a role in regulatory policies, and if there is an evidence base for this role. We will introduce these concepts in more detail below, but level of use is usually defined around a behavioural pattern of the user (see also Table 1 first row below), while potency is defined as a property or characteristic of the substance.

Level of use has long been identified as an important variable for regulatory policies, as burden of disease in general, and use disorders in particular, are linked to level of use in a dose-response fashion (e.g., see the risk curves for alcohol [10] or tobacco use [3]). In fact, heavy use over time has been suggested as a definition for substance use disorders [11] that avoids some of the pitfalls of current definitions in the DSM-5 and ICD-11 [12,13]. Potency is a term often used in pharmacology where it refers to pharmacological action (active strength) expressed in terms of the amount required to produce an effect of a given intensity [14]. In general, a highly potent drug evokes a given response at low concentrations, while a drug of lower potency evokes the same response only at higher concentrations. However, the literature clearly points out the need for more operational definitions [14,15].

Potency of substances alternatively could be defined via the internal biochemical processes they provoke, e.g., in the dopamine reward system [16]. However, such definitions have run into difficulties, as has been demonstrated in experiments in which rats given a lever choice between cocaine and sucrose administration, they reliably opt for sucrose, even if they are “cocaine addicted” [16,17].

Applying these principles to a drug like alcohol, this means that if ethanol (i.e., pure alcohol) is the active ingredient, then the ethanol concentration denotes the potency of an alcoholic beverage. Consider the onset of reduced psychomotor coordination, one of the psychoactive effects of ethanol, as the outcome, where one UK unit of alcohol (8 g ethanol) may be enough to provoke the effect for a certain body weight [18]. This effect then could result from either 25 mL of 40% whiskey or 250 mL of 4% beer or 76 mL of 13% wine [19]. These calculations assume that the effect is only based on ethanol intake and is independent of the beverage category, an assumption that is questioned and discussed below.

Potency with respect to other substances can equally be defined by the concentration of the active ingredient linked to psychoactive effects [20], i.e., nicotine for tobacco [21], opioids for opium and its natural, synthetic, and semi-synthetic derivatives, e.g., morphine, heroin, oxycodone or fentanyl (with concentration being measured in morphine equivalents [22]), and Δ^9^-tetrahydrocannabinol (THC) in cannabis [23].

The objective of this contribution is to discuss the implications of level of use and potency for the regulation of psychoactive substances. We will do this in the most detail for alcohol, but compare these policies with those for the four other most prevalent psychoactive substances, which also caused the most attributable harm [3]. Finally, we will examine whether the same principles extend to gambling. Gambling was selected here as gambling disorders have been introduced to both major classification systems in the same category as substance use disorders [12,13]. Additionally, gambling has been an object of regulation in virtually every society [24].

Thus, our analytic framework has been adopted to examine whether regulatory policies linked to different substances and gambling modalities could be derived from the same principles.

## 2. Material and Methods

In addition to studying the major textbooks on public health policy and governance for psychoactive substances and gambling [1,2,7,8,9,24,25] and WHO materials collecting survey responses from member states on policies [26,27,28,29,30,31,32], we did a search for reviews and meta-analyses of regulatory policies for the different substance categories and for gambling, and on the role of potency in regulatory policies (see Appendix A for search terms). Preferred Reporting Items for Systematic Reviews and Meta-Analyses (PRISMA) guidelines were followed [33] (see Appendix A). Results only allowed a qualitative analysis of results.

## 3. Results

There was a surprisingly small number of relevant references in the systematic search [34,35,36,37,38,39,40,41,42,43,44]. For the most part, these sources discuss high-income countries in North America and Europe, and for that reason those are the regions we focus on here.

### 3.1. Alcohol

For alcohol, the following results can be extracted. As indicated above, the main psychoactive ingredient is ethanol [45], with both short-term and long-term effects on cognition [46] and the central nervous system in general [18]. Impairment of psychomotor coordination, as described above, is one of the short-term effects [18].

However, ethanol’s effects are not restricted to its psychoactive effects; it also acts as a carcinogen [47], disturbs the innate and adaptive immune system [48], and damages multiple organs [49], contributing to more than 200 disease and injury categories [10]. Possible long-term effects of ethanol include development of alcohol use disorders or other non-communicable diseases [50]. One of ethanol’s effects is that it increases the risk of certain cancers (in part via its metabolites such as acetaldehyde), an effect over and above the carcinogenic effects of other ingredients of alcoholic beverages [10,47,51]. Usually, there are monotonic dose-response curves between level of alcohol use and disease risk [45]. Accordingly, the epidemiological indicator for heavy drinking [11] or to measure success of therapies for alcohol use disorders [52] is the level of consumption in grams of ethanol consumed per day.

Some of the regulatory policies for alcohol are based on this measure (see [26] for an overview of alcohol control policy measures by country). Taxation of alcoholic beverages is often based on grams of ethanol, in that beverages containing more ethanol are charged more tax for the same volume (so-called volumetric taxation [30,31]). Any minimum unit pricing [53] which is solely based on ethanol (i.e., not specific for beverage type), also falls into this category (e.g., the minimum unit pricing in Scotland [54], Russia [55], or in the Canadian province of Manitoba [56]). Finally, most drink-driving laws are based on ethanol concentration in blood [57]. Other alcohol control measures are independent of how much the consumer uses, such as most of the availability restrictions (e.g., opening hours for shops selling alcoholic beverages).

How does potency come into play? Alcohol policy regulations in many instances are not based on ethanol directly, but on beverage type, or bands of alcohol content as indicators of potency. First, in taxation there are beverage-specific minimum prices in some jurisdictions (e.g., in Canadian provinces [58,59]), and in many jurisdictions the tax rate depends on the beverage type, over and above the ethanol content. For example, in Ontario, Canada, the *ad valorem* tax rate is 6.1% of the beverage price for wine [58] and 61.5% for spirits [59], a 10-fold difference, when the ethanol content only varies by a factor of less than four (i.e., spirits on average 40%; wine on average 12.5% [58]). In California, the specific tax rates are 0.40 $/gallon of ethanol for beer, 0.36 $/gallon for wine, and 1.63 $/gallon for spirits [60]. In Thailand, the specific tax rates are $430 THB/litre of ethanol for beer, $1500 THB/litre for wine, and $155 THB/litre for spirits [61]. These few examples may suffice to show that while taxation often is beverage-specific, there are no clear principles that any beverage type is taxed over and above ethanol content. For example, in Thailand other considerations such as preventing initiation of alcohol use in youth by taxing the most preferred beverages for this group have played, and continue to play, a role [62].

There are also beverage-specific laws for minimum drinking age (e.g., many Western Central European countries such as Belgium, Germany, or Switzerland: 16 years for beer and wine, 18 years for spirits; all references from [26]); for other forms of availability (only sales of light beer in Sweden in corner stores; sales of beer and wine in private stores in Ontario), or for advertisement and marketing (e.g., Spain bans TV advertisements from 7:00–19:00, and in cinemas and in social media for alcoholic beverages with more than 22% ethanol only [26]. According to the Global Status Report on Alcohol Policy published in 2004, 28.6% of the countries had totally banned spirits advertisement, while 22.5% had banned wine advertisement and 15.9% had banned beer advertisement [28].

Are these alcohol policy regulations consistent with evidence that alcohol of different strengths has differential impact on harm, independent of overall ethanol consumed? First, for development of an alcohol use disorder, there is no good evidence that different beverage types have a different pathway to use disorders. Rehm and Hasan in their systematic review [63] found only weak evidence for this. Danish researchers presented a cohort study [64] where the incidence of alcohol use disorders was lower for those whose total alcohol intake was comprised of more than 35% wine, independent of the total amount of alcohol consumed. On the other hand, preferred beverage type in persons treated for alcohol use disorders seems to depend mostly on the culture [65,66]: in France in the 1970s, the majority of people with alcohol use disorders consumed predominantly wine as the beverage type as it was the main one available [67,68]; in beer-drinking countries [65,69], the majority has been shown to predominantly consume beer; and in a country where all three beverage types were almost equally available, a study found almost equal distribution of beverage preference among people with alcohol dependence [70]. Other factors include patterns of drinking and gender [63]. A prior review [71] did not focus on alcohol use disorders, but found similar results: based on North American literature, criteria used in the diagnoses of alcohol use disorders (loss of control, continued use despite harmful consequences) were mainly impacted by beer, the preferred beverage type in North America.

There is more evidence on other forms of harm related to different beverages [63,71,72]. More potent forms of alcohol, such as spirits, were associated with more acute harm caused by injury, either unintentional (e.g., traffic injury) or intentional (e.g., suicide or violence). This of course is true for spirits consumed “straight” and not as a mixed drink with non-alcoholic beverages, or for premixed drinks based on spirits. While in the underlying epidemiological research patterns of drinking were not controlled, experimental research has shown that spirits lead to higher blood ethanol concentration and psychomotor impairment, even though the overall ethanol intake is the same [63,71]. The neurobiological impairment is in part correlated with the speed of intake [18], and the same amount of ethanol can be ingested more rapidly in beverages with higher ethanol concentration.

Thus, there is some justification in the differential regulatory treatment of different beverages beyond considering the overall ethanol level [63,71,72]. This may be especially relevant for youth [73], where impact on the brain is largest [74], especially from binge drinking [75], and where unintentional and intentional injury are the leading causes of death [76]. In addition, an important regulatory strategy for alcohol seems to be to decrease the potency within beverage classes which can be achieved by progressively taxing ethanol concentration, thereby creating incentives for manufacturers to reduce the average ethanol concentration of all beverages (e.g., decreasing the average strength by 20% of each main beverage type: beer to 4%, wine to 10%; spirits to 32% [36,77]). For spirits, there is also the possibility to impose an upper limit on potency [73].

### 3.2. Other Substances and Gambling

Table 1 gives an overview of the features of different forms of psychoactive substances, their active ingredients, potency, regulations and outcomes. Obviously, we can only give specific policies for jurisdictions where a form of the substance in question is legal. Alcoholic beverages are prohibited in several countries, often for religious reasons [26]. Bhutan seems to be the only country where all tobacco products are banned from sale and distribution, although use of imported tobacco products may be legal in certain places such as hotels [78]. The non-medical use of opioids is illegal in almost all countries based on international treaties (Single Convention and the Convention on Psychotropic Substances), which most countries have ratified [79,80] although medical use is in principle allowed. Cannabis is governed by the same treaties, but some countries have chosen not to respect the treaties for this specific substance. For instance, Canada claims that their violation of international treaties for cannabis is consistent with the overarching goal of these conventions—namely, to protect the health and welfare of society [81].

Overall, for those jurisdictions with regulations and policies for at least some of the subcategories, the following can be said: (1) There are some regulations purely based on volumetric considerations, such as taxation by ethanol content, or by number of cigarettes or grams of cannabis. (2) Many of the regulations differ substantially for different potencies of the same substance class. (3) Short-term lethal consequences (overdose/poisoning) are a major consideration in regulating substance potency, especially in regions where they constitute a major cause of death (e.g., in Russia for alcohol or for opioids in North America). (4) Tobacco use is a special case, as most of the overall level of burden of disease is not linked to the psychoactive ingredient (nicotine). This has caused some controversy in regulatory handling devices and subcategories of tobacco products, where risks other than nicotine are reduced (e.g., [42]). (5) With few exceptions, cannabis use was handled internationally via prohibition [79,80], scheduling it together with substances with much higher potency and higher risk for mortality and other burden of disease (see Table 1).

Still, regulatory policies often seem to be based on risk for use disorders or other considerations, rather than on overall impact on burden of disease. It should be noted that for all substances, the psychoactive ingredients have health impacts on aother organs besides the brain (for alcohol see above, for nicotine: [43]; for opioids: [37]; for cannabis: [34,82]).

Do similar considerations apply for gambling? Like substance use, gambling has a high public health impact [24,83] and is potentially addictive [84,85], and gambling behaviour is also regulated, in many times and places by prohibition [24]. As for the addictive potential, this is defined by sets of criteria similar to those for substance use disorders, such as loss of control, increasing priority given to gambling, and continuation or escalation of gambling despite negative consequences [86].

Potency in gambling has been defined by the effect the device or activity has on the person ([87,88] see also [16]), i.e., a particular gambling modality leads to shifts of subjective experience in the gambler. And the more reliably, quickly, and robustly these shifts occur, the more potent the gambling modality becomes. Accordingly, we could define potency of gambling by the potential for average losses per minute [89]. For an overview of modalities, impact and regulations please see Table 1.

## 4. Discussion

Level of use and potency were found to be associated with regulatory policies, but not in a consistent manner. What could be the overarching principle for a more consistent approach? Often it has been argued that there should be a general principle that policies and governance should be commensurate to the overall harm [133] or at least to substance- or gambling-attributable burden of disease [1,140,141].

As an example, consider regulations for tobacco products including nicotine delivery systems that are alternatives to cigarettes, which in many jurisdictions are regulated similarly for all forms and subcategories, even though there are notable differences in harm [121,122]. As a justification for regulating alternative nicotine delivery systems similarly to cigarettes, two lines of argument are often used: first, that these systems are as addictive (i.e., leading to use disorders) as cigarettes, and second, that such systems may be the “gateway” to later cigarette smoking. Clearly, the first argument is not relevant: use disorders are only one of many different health outcomes attributable to tobacco use or alternative nicotine delivery systems, and others such as cancer or cardiovascular disease surely should and do weigh more, when a comparative statistic is used such as disability-adjusted life years [142]. The second argument may be true in certain situations, i.e., when cigarette smoking in jurisdictions is a cheaper way to obtain nicotine. For heavy users, often the cheapest way to obtain a given amount of the psychoactive substance is chosen, and this may be illegal substances, as the switch to cheaper unrecorded or surrogate alcohol in Russian people with alcohol use disorders shows ([143] for a more general overview [144,145]), or the switch from prescription opioids to cheaper street heroin or fentanyl in people with opioid use disorders in North America [146,147]. This suggests that the shift from less harmful alternative nicotine delivery systems to cigarettes could be avoided if such a shift were associated with a higher cost per unit of nicotine.

Another point to be further discussed is the overall governance of substances and gambling. As mentioned before, one reasonable principle could be that all regulations and restrictions should be commensurate to overall harm [1]. This would require a single dimension for harm, and such dimension does not exist. To give but two examples: how should the harm of a risk of criminalization for using a substance be compared to the harm of an increased risk for mortality? These harms are quite different, and there is no standard way to compare them to one another. The second example would be to compare harm to the user with harm to others (e.g., [136,148]). Would one death caused in a user (e.g., via driving under the influence of a substance) be the same as one death caused to another (e.g., killing another traffic participant under the influence of a substance)? Clearly, legislators usually prioritise harm to others over harm to the user/behaver in criminal law. While it is easy to indicate, with respect to potential harm as a criterion for control legislation, that the level of regulations should be commensurate to the level of harm, the devil will be in the details of measuring both the level of regulation and the level of harm. And expert groups doing multi-criteria analyses of harm will always be controversial (e.g., [149]).

Why are level of use and potency so relevant for regulations? From an evolutionary perspective, substance use has been part of human evolution from its early stages, but with industrial production and refined technology, previously unknown availability and potency of substances—as well as more efficient delivery mechanisms such as the hypodermic needle [150]—emerged which could not easily be coped with [1,151]. These impacts continue to the present days when it has become easy to produce high potency opioids synthetically, when potency of cannabis can be increased beyond what has been usually found in cannabis plants, or when manufacturers of EGMs have devised faster and more immersive machines [89].

Before we outline conclusions, we will note the limitations of our research. First, as in any review, we are limited by the underlying literature. Overall, there have not been that many empirical examinations of principles underlying regulatory policies and regulations for substance use and/or behavioural addictions. Second, the link between level of use, potency and health harm needs to be better explored. Third, as all substance use and gambling imply potential harm to others (see Table 1), we will need to clarify what role this harm should play in formulating regulations. For instance, regulations for driving under the influence of prescribed substances differ heavily between countries (e.g., driving under the influence of cannabis or opioids), and there seems to be no apparent reason why in one country traffic injury impacted by prescribed medication is an offense, whereas in other countries this is not the case.

## 5. Conclusions

More research is necessary on the differential effects of level of use and potency on health and other harm, but current regulatory policies for substance use and gambling are often not in line with currently available evidence of harm.

## Figures and Tables

**Table 1 ijerph-16-03749-t001:** Level of use and potency in the regulation of psychoactive substances and addictive behaviours.

	ALCOHOL	TOBACCO PRODUCTS	OPIOIDS	CANNABIS	GAMBLING
**Unit measured to define “heavy use over time” in epidemiology**	gram ethanol ( = pure alcohol)/day	cigarettes/day (as rough indicator of level of nicotine intake)	frequency of use	joints per day/week	average money spent per day [90]
**Measure of potency of psychoactive “ingredient”**	ethanol concentration	nicotine concentration; other addictive additives	opioid concentration (i.e., morphine equivalents [22])	THC concentration	potential for average losses per minute [89]
**Other impacts on the potency (psychoactive ingredient)**	Speed of delivery [18,91], food (absorption [92]; elimination [93]); tolerance; other individual factors (sex, age, body, weight [94])	Diverse factors according to delivery mode (e.g., holes for air to enter); speed of delivery [38]; individual factors	Speed of delivery, which can be manipulated by mode of administration, and which impacts on bioavailability and on onset of action [95]; tolerance [96]; other individual factors	Diverse factors, depending in part on mode of delivery; level of other cannabinoids [97] tolerance; other individual factors.	Reward parameters (e.g., jackpot size), timing parameters (e.g., speed of play, event frequency) [89,98]
**Is the level of the psychoactive ingredient a guiding principle for regulation**	In most jurisdictions, yes: taxation and drink-driving policies based on grams of ethanol [30,31] and blood alcohol content [57], respectively	Not yet in most countries, as light products are usually not differently taxed or treated by law. However, there are efforts to limit potency by law (e.g., for the US: [99,100]).	Only in part relevant for illegal opioids. Some jurisdictions have been banning (or refusing to allow) prescription opioids deemed to be high-risk due to potency but also formulation (e.g., short-acting, crushable, or otherwise easily manipulated). E.g., in 2016, Ontario de-listed several higher-strength opioids from the Ontario Drug Benefit formulary [101]. On the other hand, low potency products (e.g., codeine) are sometimes exempt from prescriptions.	Not relevant for illegal cannabis. In Canada, legal cannabis products were initially taxed per gram (i.e., no potency considerations). Dried cannabis and seeds are still taxed this way but oils are now taxed by THC content. The classes or products being permitted in 2019 (edibles, extracts and topicals) will also be taxed by quantity of THC [102].	There are regulations on 1) maximum spend per hour, day (e.g., [24] for the Netherlands), with sometimes distinctions on where the gambling takes place; or on the 2) maximum that can be spent by bet (e.g., the UK implemented, as of April 2019, £2 bet limits on certain EGMs (more or less equivalent to slot machines [103])). In additions, restrictions based on location are possible based on potency (e.g., high potency for casinos only [24]).
**Can direct short-term effects be lethal?**	Yes, in some jurisdiction alcohol poisoning is a major cause of death (e.g., Russia [104]).	Yes, but such cases are rare [105].	Yes, in some jurisdictions opioid overdose is a major cause of death (USA [106]; Canada [107]).	No, not with plant-based cannabis. However, synthetic cannabinoids can be lethal [108].	No
**Are there different subcategories of psychoactive substances/devices?**	Yes: beer, cider, mixed drinks, wine, fortified wine, spirits (e.g., for a distribution of main beverage types by country, region, and globally [26]).	Yes: the Tobacco Framework Convention defines tobacco products as “products entirely or partly made of the leaf tobacco as raw material which are manufactured to be used for smoking, sucking, chewing or snuffing” [109]; new products considered are “Heated tobacco products” or “electronic nicotine delivery systems”, the latter not a tobacco product [110].	Yes: based on two dimensions. First, there are illegal and medicinal products. Second, there are different categories of opioids such as codeine, fentanyl, heroin, morphine.	Yes: based on two dimensions. First, there are illegal and medicinal products; and in select jurisdictions legal recreational products. Second, there are different categories of cannabis such as smokable or edible products	Yes: e.g., casino gambling without electronic gambling machines (EGMs); EGMs; lotteries; Internet gambling; horse racing and other betting on animal contests [111]. Regulation is mainly concerned with commercial gambling.
**Do subcategories differ in potency?**	Yes	Not necessarily; some of the electronic nicotine delivery systems have higher potency [112].	Different opioids (e.g., fentanyl, heroin, codeine) differ in potency; however, the distinction between illegal and medicinal products is not relevant, as both subcategories can have different potencies.	Both illegal and medicinal products can have different potencies, so this distinction is not relevant for potency; however, different plants and types of cannabis differ in potency. In addition, synthetic cannabinoids can have higher potency [113].	Yes
**Evidence for justifying different regulations for different subcategories based on addictive potential**	For causing alcohol use disorders: weak evidence for differentiating subcategories [63].	All product classes may lead to tobacco use disorders based on nicotine concentration and the device used.	All categories of opioids may lead to opioid use disorders, depending on different factors including potency [114,115,116].	Both illegal and medicinal cannabis may lead to cannabis use disorders, depending on different factors including potency [117,118].	Yes, electronic gambling machines were associated with higher risk of gambling addiction as defined by ICD ([89,119] but [120])
**Evidence for justifying different regulations for different subcategories based on overall harm**	More potent forms of alcohol such as spirits were associated with more unintentional and intentional injury [63,71,72]. While in the underlying epidemiological research, patterns of drinking often were not controlled for, experimental research also showed that spirits led to higher blood ethanol concentration and psychomotor impairment, even if the overall ethanol intake was the same [63,71].	There are differences in overall harm of different tobacco products and of electronic nicotine delivery systems [121,122]. The reason for these differences can be explained by the fact that many other ingredients of cigarettes and other tobacco products over and above nicotine impact on morbidity and mortality [123]. In other words, while nicotine has been linked to morbidity and mortality [123,124,125,126], even cigarettes without nicotine would have markedly detrimental health effects.	Health harm, especially overdoses, seems to be mainly based on dose, which is linked to potency ([37], but see [127]), not on the distinction between medicinal and other products. The recent fentanyl crisis in North America certainly has been linked to high potency of this opioid, its synthetic analogues and other synthetic opioids [128,129]. Obviously, legal problems as a form of harm by definition only apply to illegal products.	No indication that medicinal products have less health harm; obviously, legal problems as a form of harm are only relevant for illegal products. Synthetic cannabinoids may cause higher harm [113]. Overall, while cannabis use can cause health harm [82], the level of this burden is much lower than for other substances [3,130]), even when considering prevalence of use [131].	Certain structural characteristics of EGMs have been identified as strongly associated with harm, including high speed of play, losses disguised as wins, near misses, and features that give the user the illusion of control [132].
**Evidence on harm to others (see also** [133] **as overview of harm to others from substance use)**	Considerable harm to others (highest of all substances in the comparative study of [134]), in part linked to patterns of drinking (traffic injury, FASD; [135]) and maybe to potency (violence and other injury [63,71]).	Considerable health harm to others, mainly known for cigarette smoking [136].	Considerable harm to others via needle sharing (infectious disease; [137] and traffic injury [138]).	Harm to others via traffic injury [139].	Yes, but lesser degree than substance use, mainly to families and immediate environment, and no fatal outcomes [24]

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
