# Peer review of "Regulatory Policies for Alcohol, other Psychoactive Substances and Addictive Behaviours: The Role of Level of Use and Potency. A Systematic Review"

_ijerph, 2019, doi:10.3390/ijerph16193749_

Round 1

Reviewer 1 Report

This article was an original systematic review that examines the degree to which the level of use and potency play a role in informing the regulatory policies.
The work demonstrates an excellent ability to deal with complexity, contradictions and incomplete information.
It was good to have the materials and method section and the combined supplementary material which enabled the reader to understand where the authors obtained their data and what approach they took.
It was a very confidently argued piece of research, with clear justifications.

Author Response

This article was an original systematic review that examines the degree to which the level of use and potency play a role in informing the regulatory policies.
The work demonstrates an excellent ability to deal with complexity, contradictions and incomplete information. 
It was good to have the materials and method section and the combined supplementary material which enabled the reader to understand where the authors obtained their data and what approach they took.
It was a very confidently argued piece of research, with clear justifications.

Thanks!

Reviewer 2 Report

The manuscript concerns a very significant problem in terms of physical and mental health, functioning in society as well as the social harm of addiction. The manuscript has been properly formatted, divided into parts in accordance with the requirements of the journal. It is written in an interesting, factual manner and has correctly selected and up-to-date literature. However, it contains a few weak points, which the Authors also mentioned in the discussion.

The first and most important matter is the title of manuscript and choice of the described substances and behaviors. I think that gambling does not match to the substances chosen by the Authors. They should rather focus only on psychoactive substances. If, nevertheless, They want to expand the range of assessed risk behaviors, then the impact of internet addiction, shopping addiction, overeating, etc. should also be discussed.

If the Authors will focus only on addictive substances, They should discuss also a problem of overusing and regulatory policies for substances such as benzodiazepines, ephedrine, methylphenidate or dextromethorphan, since it is becoming a growing problem today.

The font should be standardized. The Authors claimed that even cocaine addicted rats opt for sucrose when they have a choice between cocaine (synthetic reward) and sucrose (natural reward). However, this is probably an isolated case since the synthetic rewards are always more addictive and chosen by animals more willingly than natural. This topic should be discussed in more details. The Authors should indicate exactly which area (countries) the research concerns. If in table 1 the Authors indicated that exist subcategories differ in potency in the subject of gambling, then these subcategories should be listed and the references should be provided. In table 1 in evidence on harm to others, the social and psychological effects of addiction for co-addicts should be clearly stated. It would be worthwhile to provide the impact of psychoactive substances on different organs in table and then the harmfulness of these substance could be compared. It is not enough to mention the references including this subject as it is in the current form of the manuscript. Finally, it is worth pointing out that when preparing the next manuscript the Authors should focus on the limitations of current work which They mentioned in discussion.

Author Response

The manuscript concerns a very significant problem in terms of physical and mental health, functioning in society as well as the social harm of addiction. The manuscript has been properly formatted, divided into parts in accordance with the requirements of the journal. It is written in an interesting, factual manner and has correctly selected and up-to-date literature. However, it contains a few weak points, which the Authors also mentioned in the discussion.

Thanks!  We will answer in a point by point below.

The first and most important matter is the title of manuscript and choice of the described substances and behaviors. I think that gambling does not match to the substances chosen by the Authors. They should rather focus only on psychoactive substances. If, nevertheless, They want to expand the range of assessed risk behaviors, then the impact of internet addiction, shopping addiction, overeating, etc. should also be discussed.

Gambling has been introduced into both major diagnostic systems as the prototypical behavioural addiction, and thus is of special interest to be compared with psychoactive substances.  This background has been added to the text.

Moreover, gambling has been regulated in most jurisdictions, whereas internet or shopping addiction or overeating have not.

American Psychiatric Association. (2013). Diagnostic and statistical manual of mental disorders: DSM-5 (5th ed.). doi: 10.1176/appi.books.9780890425596

World Health Organization. (2016). International statistical classification of diseases and related health problems (10th Revision, Version: 2016). Retrieved from https://icd.who.int/browse10/2016/en

World Health Organization. (2018). International statistical classification of diseases and related health problems (11th Revision). Retrieved from https://icd.who.int/browse11/l-m/en

If the Authors will focus only on addictive substances, They should discuss also a problem of overusing and regulatory policies for substances such as benzodiazepines, ephedrine, methylphenidate or dextromethorphan, since it is becoming a growing problem today.

We wanted to concentrate on the most prevalent substance use disorders, and thus restricted ourselves to the current list.

Degenhardt, L., Charlson, F., Ferrari, A., Santomauro, D., Erskine, H., Mantilla-Herrara, A., Whiteford, H., Leung, J., Naghavi, M., Griswold, M., Rehm, J., Hall, W., Sartorius, B., Scott, J., Vollset, S.E., Knudsen, A.K., Haro, J.M., Patton, G., Kopec, J., Carvalho Malta, D., Topor-Madry, R., McGrath, J., Haagsma, J., Allebeck, P., Phillips, M., Salomon, J., Hay, S., Foreman, K., Lim, S., Mokdad, A., Smith, M., Gakidou, E., Murray, C., & Vos, T. GBD 2016 Alcohol and Drug Use Collaborators. (2018).  The global burden of disease attributable to alcohol and drug use in 195 countries and territories, 1990-2016: a systematic analysis for the Global Burden of Disease Study 2016.  Lancet Psychiatry, 5(12), 987-1012.  doi: 10.1016/S2215-0366(18)30337-7 

The font should be standardized.

We made sure the font has been standardized throughout the text and the references.  The font for the Table is different and it is up to the journal to change this as they see fit.

The Authors claimed that even cocaine addicted rats opt for sucrose when they have a choice between cocaine (synthetic reward) and sucrose (natural reward). However, this is probably an isolated case since the synthetic rewards are always more addictive and chosen by animals more willingly than natural. This topic should be discussed in more details.

We could not find any such generalizations, but the literature on cocaine vs. sugar as reward is clear and referenced.

The Authors should indicate exactly which area (countries) the research concerns. If in table 1 the Authors indicated that exist subcategories differ in potency in the subject of gambling, then these subcategories should be listed and the references should be provided. In table 1 in evidence on harm to others, the social and psychological effects of addiction for co-addicts should be clearly stated. It would be worthwhile to provide the impact of psychoactive substances on different organs in table and then the harmfulness of these substance could be compared. It is not enough to mention the references including this subject as it is in the current form of the manuscript. Finally, it is worth pointing out that when preparing the next manuscript the Authors should focus on the limitations of current work which They mentioned in discussion. 

We now mentioned that most of our examples stem from high income countries, and that may be a limitation.

Reviewer 3 Report

The article entitled “Regulatory policies for alcohol, other psychoactive substances and addictive behaviors: the role of level of use and potency. A systematic review " is interesting in the context of broadening the reader's knowledge of the legal aspects of regulating the use of psychoactive substances and gambling. The text, based on a rich review of the literature, contains valuable analyzes concerning the role of the level of psychoactive substance use and their impact. They can be useful in the practice of addiction therapy work in the context of learning about factors influencing the process of both chemical and behavioral addiction. The work supplements the current deficiencies in the literature in this field. The substantive and methodological content  presents a very good level of preparation.

However, it is worth introducing several modifications before publication:

suggests that the authors introduce, as an introduction to the discussed issues of legal regulations of alcohol consumption, tobacco, opioids, cannabis and gambling regulations an extended analysis of the concepts (including analysis of differences) of addiction to psychoactive substances and behavioral addictions. The current record, in the opinion of the reviewer, is insufficient. The authors cite only a reference to the addiction definitions  of  DSM-V and ICD-11 without further discussion in the context of their analyzes.

 I recommend referring to the situation of cross-dependence and their presence in regulatory policies when describing addiction to both chemical substances and behavioral addictions

it is worth clarifying the regulatory policies of which countries or regions of the world were subject to analysis in the context of legal addiction regulations. Reading the text to date means that the focus of regulatory policy analysis is transferred to the countries of North America and Central and Western Europe. It is worth or ensuring balance in recalling the text with examples also from other parts of the world (South America, Africa, Australia) or clearly stating that the text does not apply to them and for what reason.

it is worth broadening the analysis of the cultural drinking pattern

it is worth clarifying the title of the article so that the reader knows specifically that selected (i.e. not all) psychoactive substances are subject to analysis. In its current form, the title may be misleading.

order the numbering of individual parts of the article should to be corrected. The authors in section Results distinguish point 1.1 Alcohol. However, there is no point 1.2 or 1.3. The reviewer presumes that the subsection Other substances and gambling is section 1.2. If we divide something, it is divided into at least two parts.

The text, after making little corrections, is worth publishing, which I recommend.

Author Response

The article entitled “Regulatory policies for alcohol, other psychoactive substances and addictive behaviors: the role of level of use and potency. A systematic review " is interesting in the context of broadening the reader's knowledge of the legal aspects of regulating the use of psychoactive substances and gambling. The text, based on a rich review of the literature, contains valuable analyzes concerning the role of the level of psychoactive substance use and their impact. They can be useful in the practice of addiction therapy work in the context of learning about factors influencing the process of both chemical and behavioral addiction. The work supplements the current deficiencies in the literature in this field. The substantive and methodological content presents a very good level of preparation.

Thanks!

However, it is worth introducing several modifications before publication:

suggests that the authors introduce, as an introduction to the discussed issues of legal regulations of alcohol consumption, tobacco, opioids, cannabis and gambling regulations an extended analysis of the concepts (including analysis of differences) of addiction to psychoactive substances and behavioral addictions. The current record, in the opinion of the reviewer, is insufficient. The authors cite only a reference to the addiction definitions of DSM-V and ICD-11 without further discussion in the context of their analyzes.

We now give some more introduction as to why we included gambling in this analysis.

I recommend referring to the situation of cross-dependence and their presence in regulatory policies when describing addiction to both chemical substances and behavioral addictions

While co-dependence or cross-dependence exist and may become a clinical challenge, for the current regulatory framework, we wanted to restrict ourselves to the elaboration of use disorders of one substance and gambling.

it is worth clarifying the regulatory policies of which countries or regions of the world were subject to analysis in the context of legal addiction regulations. Reading the text to date means that the focus of regulatory policy analysis is transferred to the countries of North America and Central and Western Europe. It is worth or ensuring balance in recalling the text with examples also from other parts of the world (South America, Africa, Australia) or clearly stating that the text does not apply to them and for what reason.

In the revision we gave reason for the examples chosen (they reflect the examples given in our sources).

it is worth broadening the analysis of the cultural drinking pattern

While patterns of drinking are important for alcohol-attributable harm, and pattern of use in general maybe important for attributable harm of other substances as well, our analyses was concentrated on level of use and potency in their potential impact on regulations.  Another dimension would have resulted in a different paper.

it is worth clarifying the title of the article so that the reader knows specifically that selected (i.e. not all) psychoactive substances are subject to analysis. In its current form, the title may be misleading.

 We chose to keep the current title as it reflects the text without promising to discuss all psychoactive substances:

Regulatory policies for alcohol, other psychoactive substances and addictive behaviours: the role of level of use and potency.  A systematic review

Moreover, we give reasons why the substances were selected in the revised text.

order the numbering of individual parts of the article should to be corrected. The authors in section Results distinguish point 1.1 Alcohol. However, there is no point 1.2 or 1.3. The reviewer presumes that the subsection Other substances and gambling is section 1.2. If we divide something, it is divided into at least two parts.

We changed accordingly and numerated section 1.2

The text, after making little corrections, is worth publishing, which I recommend.

Thank you!